# The demographics and geographic distribution of laboratory-confirmed Lyme disease cases in England and Wales (2013–2016): an ecological study

John S P Tulloch,[1,2] Amanda E Semper,[3,4] Tim J G Brooks,[3,4] Katherine Russell,[5] Kate D Halsby,[5] Robert M Christley,[1,6] Alan D Radford,[1,6] Roberto Vivancos,[2,7] Jenny C Warner[3,4]

**Correspondence to**
Dr John S P Tulloch;
jtulloch@liverpool.ac.uk

## ABSTRACT

**Objective** Lyme disease is a tick-borne disease of increasing incidence and public concern across the Northern Hemisphere. However, the socio-demographics and geographic distribution of the population affected in England and Wales are poorly understood. Therefore, the proposed study was designed to describe the demographics and distribution of laboratory-confirmed cases of Lyme disease from a national testing laboratory.

**Design** An ecological study of routinely collected laboratory surveillance data.

**Setting** Public Health England's national Lyme disease testing laboratory.

**Participants** 3986 laboratory-confirmed cases of Lyme disease between 2013 and 2016.

**Results** In England and Wales, the incidence of laboratory-confirmed Lyme disease rose significantly over the study period from 1.62 cases per 100 000 in 2013 to 1.95 cases per 100 000 in 2016. There was a bimodal age distribution (with peaks at 6–10 and 61–65 years age bands) with a predominance of male patients. A significant clustering of areas with high Lyme disease incidence was located in southern England. An association was found between disease incidence and socioeconomic status, based on the patient's resident postcode, with more cases found in less deprived areas. Cases were disproportionately found in rural areas compared with the national population distribution.

**Conclusions** These results suggest that Lyme disease patients originate from areas with higher socioeconomic status and disproportionately in rural areas. Identification of the Lyme disease hotspots in southern England, alongside the socio-demographics described, will enable a targeted approach to public health interventions and messages.

## INTRODUCTION

Lyme disease is an important zoonotic tick-borne disease caused by spirochaetes of the *Borrelia burgdorferi sensu lato* genospecies complex. It is spread through the bites of infected *Ixodes* ticks, in the UK, primarily *Ixodes ricinus*.[1] Autochthonous cases are

## Strengths and limitations of this study

► This study is based on a national testing laboratory's figures and provides a much needed update on basic epidemiological information about Lyme disease in England and Wales.
► Data on the socio-economic status of Lyme disease cases are globally sparse; our findings will have implications for future public health awareness and intervention schemes and may offer new avenues for research.
► Lyme disease incidence maps have been produced to a high resolution and show significant clustering of disease, providing public health organisations with locales to target interventions.
► Geographical data, and associated variables, were based on patient residence information rather than tick bite location.
► The study was of an ecological design and positive cases were compared with the national population; therefore, no measures of risk or multivariable analysis of demographic variables were possible.

found solely in the Northern Hemisphere.[2 3] Most commonly, early infection presents with an erythaema migrans rash, with associated generalised flu-like symptoms.[4] Neurological manifestations, such as facial nerve palsy, can occur as part of early disseminated infection.[2] The varied presentation of the disease and the potential of increased tick exposure risk due to the extension of tick habitats as a result of changes in land management, climate and human activity have resulted in heightened awareness and surveillance by public health organisations.[5 6]

In Western Europe, the population-weighted incidence has been estimated at 22.04 cases per 100 000 person years.[7] In the UK, Lyme disease is not a notifiable disease, but laboratory-confirmed *Borrelia* spp. are notifiable causative organisms.[8] Public Health

England (PHE) compiles data on laboratory-confirmed cases of Lyme disease in England and Wales, which show a rise in the national incidence of confirmed cases from 0.38 per 100 000 population in 1997[9] to 1.95 per 100 000 population in 2016.[10] Data on laboratory-confirmed cases are provided by the national diagnostic laboratory, the PHE Rare and Imported Pathogens Laboratory (RIPL), which provides specialist advice and diagnostics for Lyme disease to the National Health Service (NHS) in England and Wales. Laboratory testing is based on serological diagnosis using a combination of screening and confirmatory immunoassays in accordance with internationally accepted best practice for Lyme disease diagnosis.[4 11 12] The incidence of cases which does not require laboratory diagnostics is unknown. These cases are most likely presented to and are clinically diagnosed and managed solely within primary care, as recommended by The National Institute for Health and Care Excellence (NICE) guidelines.[4] The incidence of Lyme disease cases seen in primary care in the UK has been estimated to be between two and six times the number confirmed by laboratories.[13]

Information regarding the demographics of Lyme disease cases in England and Wales is limited. Laboratory surveillance data published in 2000 describe an equal sex ratio at all ages; however, numbers were not provided and statistical comparison was not performed.[14] They describe a bimodal age distribution with peaks in childhood and at 45–64 years old. Hospital admissions data investigating Lyme disease and Bell's palsy describe a similar bimodal distribution.[15] These findings are similar to other European countries.[16–18] There is a sparsity of recent demographic data for Lyme disease in England and Wales. The geographic distribution of confirmed cases was last described in 2000.[14] They describe a tendency for cases in southern England, especially around the New Forest. However, this data may not reflect the current distribution of Lyme disease cases in England and Wales. More current data are urgently needed to enable targeted public health messaging and intervention strategies.

Globally, the negative income and education gradient of health have helped shape public health strategy and policy.[19 20] As a person's position on the socioeconomic spectrum increases, so their likelihood of better health increases. Such potentially avoidable disparities in health has led to an increased focus on understanding the social determinants of health[21] and developing measures to address these. Work to explore the association between socioeconomic status and Lyme disease incidence is limited. In the USA persons were found to be at greatest risk of Lyme disease if they lived in the highest or lowest socially vulnerable areas.[22] Two studies found a relationship between Lyme disease incidence and median annual household income, with incidence peaking at around US$ 80,000.[23 24] However, a consistent relationship between the socioeconomic state of an individual and their Lyme disease acquisition risk has yet to emerge. In particular, no in-depth research has been performed in Europe investigating the socioeconomics of the Lyme disease patient cohort.

The aim of this study was to use information collected through routine surveillance in England and Wales to describe the demographics and geographic distribution of laboratory-confirmed Lyme disease cases over a 4-year period. Correlations between Lyme disease incidence and socioeconomic indices were analysed, using patient residence postcode as a proxy for individual patient characteristics. New insight will be provided into the key demographic, geographical and social determinants of the Lyme disease patient population. This would allow us to identify potentially at-risk populations, shape public health interventions and assist in appropriate disease awareness.

## METHODS

A retrospective analysis was performed using data extracted from the PHE RIPL laboratory information management system (LIMS), between 1 January 2013 and 31 December 2016, for laboratory-confirmed Lyme disease cases, the same data as used for PHE's zoonoses report.[25] The RIPL LIMS contains information provided on the Lyme disease referral form submitted at the time of sample submission and any additional information provided by clinicians during case follow-up and management.[26] The form captures information on the age, gender, location, clinical symptoms and travel history of the patient. Data were cleaned and duplicate (across all variables) records were removed where necessary.

Annual Lyme disease incidence estimates were calculated, using the Office for National Statistics (ONS) mid-year population estimates as the denominator population.[27] A $\chi^2$ test for trend and a $\chi^2$ test for departure from the trend were used to analyse trends in incidence. Cases were stratified by age and gender. Using binomial tests, the null hypothesis that there was no difference in case numbers between males and females was tested within differing age bands, and overall.

Geographical information was collated based on (1) the regional origin of a diagnostic sample (usually a hospital microbiology department) consisting of eight PHE regions, and Wales as a whole,[28] and (2) the postcode area of the patient. These were used to calculate average annual incidence for the study period. To account for the unknown distance between a patient's home address and where they were bitten and to highlight any disease hotspots, the disease incidence map for postcode area was smoothed. A k-nearest neighbours (k-NN) approach was used.[29–31] In this approach, a Queen contiguity was used to define geographical neighbours; this defines a neighbour as being an area that shares a common edge or vertex. k is defined as the number of neighbours used for smoothing. k is equal to the square root of the total number of discrete geographical areas rounded to the nearest whole odd number (ie, 105 postcode areas, its square root being 10.2; therefore k=11). Exploratory spatial data analysis

(ESDA)[32 33] was used to explore the spatial autocorrelation of the postcode area incidence map. Global and local Moran's I values were calculated, and a local indicators of spatial association (LISA) significance map constructed to highlight any significant clusters. In both the k-NN smoothing and Moran's I calculations, a queen adjacency matrix was used.

Patient postcode was linked to ONS socioeconomic data,[27] enabling a description of the socioeconomic characteristics of the population in which a Lyme disease case was resident. If no patient postcode was recorded, these cases were excluded from the analysis. Socioeconomic status is reported through the English indices of deprivation (EID) 2015[34] and the Welsh Index of Multiple Deprivation (WIMD) 2014[35] (online supplementary material 1, table 1). Postcode area case count data were matched independently to the EID and WIMD, and a rural urban classification. As EID and WIMD are on a discrete ordinal scale, Spearman's rank correlation was used to calculate the correlation between the number of cases and deprivation score. The proportion of cases with their home addresses located in either a rural or urban area was compared with the national rural urban classification from the ONS.[36] This was performed using a $\chi^2$ test of independence for both English and Welsh data.

All statistical and spatial analyses were carried out using R language (V.3.2.0) (R Core Team 2015). Results were deemed significant where $p < 0.05$.

## Patient and public involvement

The public or patients were not involved in the development of the research question or the outcome measures. However, this research was informed by the research recommendations in the 2018 Lyme disease NICE guidelines,[4] which had patient and public involvement. Investigators have and will continue to present these findings at regional and national events and to the general public, patients groups, NHS organisations, public health departments and governments agencies.

## RESULTS

In total 3986 unique cases, 3893 cases in England and 93 in Wales, meeting a serological diagnosis of Lyme disease, were identified in the RIPL LIMS between 1 January 2013 and 31 December 2016. Of these, 98.7% (n=3935) had complete records for date of submission, gender and age. Only 10.5% (n=417) of cases had details on the submission form, confirming or excluding international travel from a case's clinical history. Due to the low completeness of this variable, it was concluded that further analysis of travel history would not be performed.

The annual incidence of laboratory-confirmed Lyme disease cases in England and Wales rose from 1.62 per 100 000 population in 2013, to 1.95 in 2016. These figures are identical to PHE's official incidence figures as they used the same data source.[10] There was evidence of an overall association between incidence and year ($\chi^2$=43.13,

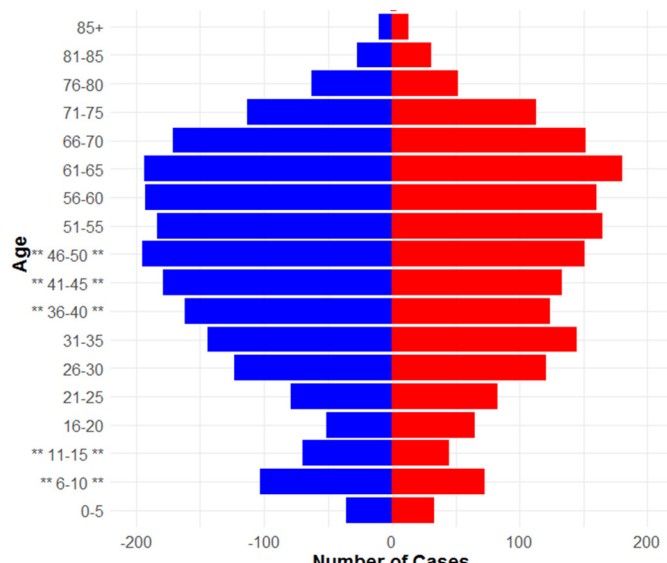

**Figure 1** Population demographics of laboratory-confirmed Lyme disease cases in England and Wales, 2013 – 2016. (Asterisks represent age bands with a significant difference between genders. Male = blue, female = red).

$p < 0.001$). This association took the form of a trend with increasing incidence each year ($\chi^2$=30.17, $p < 0.001$). Departures from the trend were significant ($\chi^2$=43.1–30.1=12.96, $p < 0.001$), as shown by the fall in incidence in 2014. There was marked seasonality, with the peak numbers of cases being diagnosed in the summer months each year (figure 1).

Across all ages there were significantly more male (n=2096) than female (n=1839) cases ($p < 0.001$), with a bimodal age distribution, with peaks at 6–10 and 61–65 year age bands (figure 2). Grouping the data in 5-year age bands, there were significantly more men than women in the 6–10 (p=0.03), 11–15 (p=0.03), 36–40 (p=0.01), 41–45 (p=0.02) and 46–50 (p=0.04) age groups.

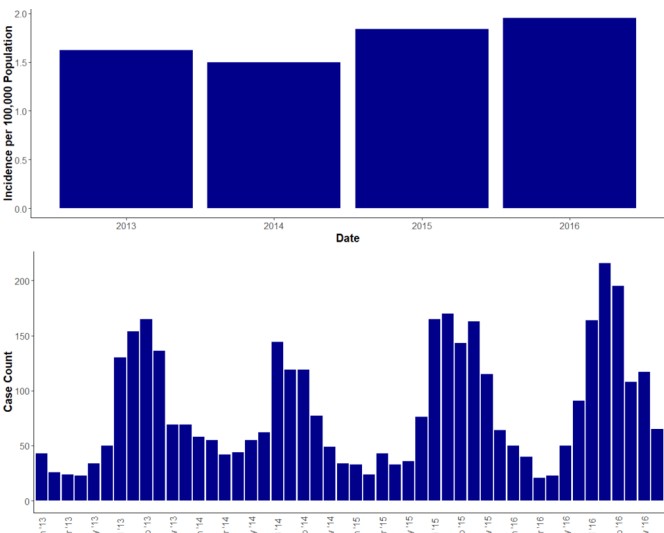

**Figure 2** The annual incidence of Lyme disease in England and Wales (2013 – 2016), and the number of cases per month

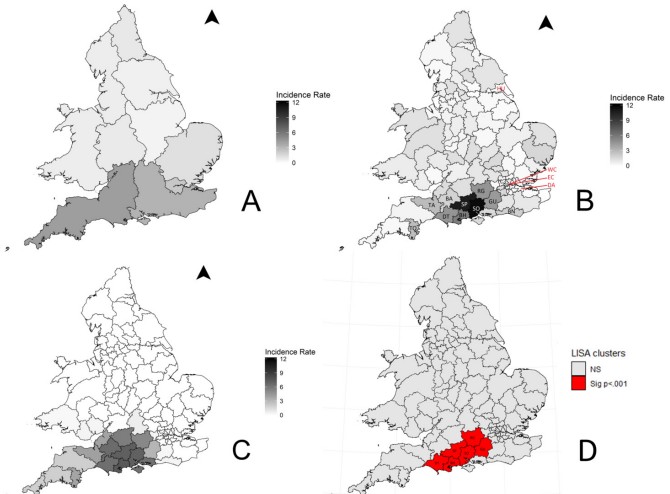

**Figure 3** The average incidence (cases per 100 000 per year) of laboratory-confirmed Lyme disease in England and Wales (2013–16) ((A) Public Health England region and Wales (n=3985), (B) Patient postcode area (n=2321), (C) Smoothed patient postcode area, (D) LISA map of significant incidence clusters. Highest postcode areas and clusters are labelled accordingly; SO—Southampton, SP—Salisbury, BH—Bournemouth, RG—Reading, DT—Dorchester, GU—Guildford, TA—Taunton, TQ—Torquay, BN—Brighton and BA—Bath. Areas with no cases are labelled in red; DA—Dartford, EC—Eastern Central London, HU—Hull and WC—Western Central London).

Data were available about PHE regions for 99.9% (n=3985) of the study population (figure 3A). The patient residence postcode was not provided on 1665 of the referral forms, and therefore only 58.2% (n=2321) of cases could be described at postcode area resolution. The average percentage of missing postcode data by PHE region was 31.9% (range: 10.8%–76.1%). The regions with the highest missing postcode data were London (76.1%), South West (49.4%) and North West (44.7%). The regions with the lowest missing postcode data were Wales (10.8%), North East (12.1%) and West Midlands (14.5%). The South West PHE region had the highest incidence of Lyme disease in England and Wales; none of the PHE regions, nor Wales, reported zero cases. The postcode areas with the highest average annual incidence of Lyme disease were Southampton (11.65 cases per 100 000 per year), Salisbury (10.75), Bournemouth (5.62), Reading (4.59), Dorchester (4.57), Guildford (4.31), Taunton (2.79), Torquay (2.75), Brighton (1.96) and Bath (1.84) (figure 3B). These areas are all in southern England. Only four postcode areas had no laboratory-confirmed cases in the 4-year surveillance period (figure 3B), namely Dartford, Eastern Central London, Hull and Western Central London. The smoothed data showed a trend for the areas of highest incidence to be located in southern-central England (figure 3C). There was significant spatial autocorrelation, the global Moran's I was 0.564 (p=0.01), indicating that postcode areas with similar incidence are clustered together. LISA mapping identified six areas as significant clusters of high incidence

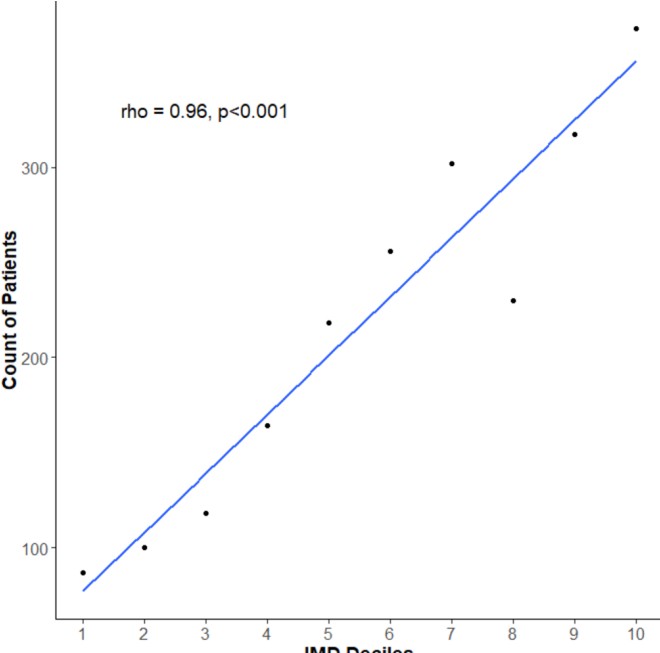

**Figure 4** Relationship between laboratory-confirmed Lyme disease case numbers (2013–2016) in England and the English indices of deprivation 2015.

(figure 3D); Southampton, Salisbury, Bournemouth, Reading, Dorchester and Guildford (for all p<0.001).

Using patient residence postcode data, it was possible to match 55.6% (n=2165) of English records to the EID and 98.2% (n=92) of Welsh records to the WIMD. An overall significant positive correlation between the number of cases and Index of Multiple Deprivation (IMD) decile was observed (ρ=0.96, p<0.001), with more Lyme disease cases found in less deprived areas (figure 4). This significant positive correlation was seen across all domains of deprivation, except the 'Barriers to Housing and Services Domain' where this trend was reversed (ρ=−0.88, p=0.002) and the 'Living Environment Deprivation Domain' where there was no significant correlation (ρ=0.2, p=0.58) (online supplementary material 2). An overall significant positive correlation between the number of cases and WIMD rank was observed (ρ=0.89, p=0.04), with more Lyme disease cases found in the least deprived areas.

When compared with the national population, the study population was disproportionately more likely to live in a rural area, for both English (p<0.001) and Welsh (p<0.001) sections of the study population (table 1).

**Table 1** The rural urban classification of laboratory-confirmed cases of Lyme disease in England and Wales (2013–2016) compared with the national census population

| Category | Percentage of English study population | Percentage of Welsh study population | Percentage of 2015 census population |
|---|---|---|---|
| Rural | 34.3% (n=743) | 47.8% (n=44) | 17.9% |
| Urban | 65.7% (n=1422) | 52.2% (n=48) | 82.1% |

## DISCUSSION

Between 2013 and 2016 there was a significant increase in the annual incidence of laboratory-confirmed Lyme disease cases, with a seasonality that matched previous publications and has been well documented.[9] The observed seasonality closely matches *I. ricinus* tick population dynamics in the UK, which annually peak around June and July.[1 37] Concerns have been raised about how the expansion of tick habitats due to changes in land use and management, and climate change, may be increasing the risk of Lyme disease infection.[5 38] Although the incidence of confirmed cases increased over the study period, there was significant deviation from the trend, most notably in 2014. The reasons behind this variable, but increasing, incidence of Lyme disease are likely to be multifactorial and may include raised public and practitioner awareness, variable weather patterns causing alterations in tick abundance and/or carriage of *B. burgdorferi s.l.,* and changes in human activity and behaviour.

This study observed a bimodal age distribution, with peaks at 6–10 and 61–65 years, and an overall predominance of males. This bimodal distribution has been reported in other European countries,[16–18] and matches previous UK studies.[14 15] However, the predominance of males in the current study population does not concur with other European studies, where women are over-represented.[16–18] In the USA, Lyme disease is more prevalent in males compared with females less than 60 years old, and equal or higher in women above 60 than among men.[2] In contrast, more men were hospitalised in France due to Lyme disease and more women were diagnosed by general practitioners.[39] Historically, in England and Wales, Lyme disease incidence in men and women has been similar.[14 15] The male predominance in this study may be due to the difference in health seeking behaviour between genders, with women more likely to seek healthcare at early stages of illness.[40] By presenting at later stages of Lyme disease, when pathognomonic signs may have waned, male cases may require laboratory confirmation more frequently. Further work is needed to establish the causes behind these gender differences and whether they are related to environmental or behavioural risk factors such as occupation, leisure activities or differences in health seeking behaviours.

There was geographical variation in Lyme disease incidence across patient residence postcode area in England and Wales, based on 58.2% of laboratory-confirmed cases. The global Moran's I statistic showed that there was significant positive spatial autocorrelation, and clusters of high incidence were found in southern England. This area includes the New Forest National Park, the South Downs National Park, Salisbury Plain, Cranborne Chase Area of Outstanding Natural Beauty (AONB), Dorset AONB and Purbeck Heritage Coast. These are all popular destinations for outdoor activities and are in southern England where the Lyme disease vector *I. ricinus* is most prevalent.[1 5 41] The exposure risk from ticks is likely to be higher in these areas than other parts of the country.

It is interesting that previously observed Lyme disease hotspots, such as Thetford Forest,[14] were not evident in the current study. This may be due to changing tick population dynamics and/or the prevalence of *B. burgdorferi s.l.* infection in host-seeking vectors, changing human behaviour, or the larger number of patients within this study population. It is also possible that awareness of Lyme disease is higher in these areas, and cases are successfully identified and managed in primary care without the need for serological diagnosis. Throughout the rest of England and Wales the incidence of confirmed Lyme disease cases remains relatively low (69.2% of resident postcode areas have an incidence of less than 1.0 per 100 000 population per year) compared with the majority of Western Europe.[7] The four postcode areas with no laboratory-confirmed cases were all surrounded by areas with very low incidence and are likely to be reflective of the overall low incidence of Lyme disease in England and Wales. Although *I. ricinus* ticks are widespread across England and Wales,[1] the risk of contracting Lyme disease appears to be relatively low. It is possible that the tick populations found within high Lyme disease incidence areas may also have the highest *B. burgdorferi s.l.* prevalence. Several studies would appear to support this hypothesis,[42–44] but further work is needed to compare the incidence of human cases, abundance of ticks and prevalence of *B. burgdorferi s.l.* in ticks in the same geographic area. The areas with high incidence are predominantly rural and this is reflected in the results where the study population were disproportionately more rural compared with the national population. Information about case locality represented by PHE region is reflective of the case's referring hospital microbiology department rather than the cases' residence, or location of exposure. In some instances, mainly in rural areas, this hospital may be a significant distance from the abode of the patient. This figure therefore is more reflective of the burden of Lyme disease on local microbiology departments.

Information provided at postcode area level relates to the patient's home address, and not necessarily to where the patient was bitten by a tick. Some patients are likely to have been bitten outside their resident postcode area. The further the exposure from home, the larger this spatial error will be. To date, no work has been done to quantify this error in the UK. The smoothed map (figure 3C) attempts to account for this and shows an area of high incidence in southern-central England, centred around Southampton, Salisbury and Weymouth and extends further west than the raw incidence data. This map highlights theoretical Lyme disease risk areas more accurately, as it accounts for the bite distance spatial error, and should be the map used for targeting public health strategies. The observed strong geographical clustering of positive cases (figure 3D) suggests that patient residence postcode does correlate to some extent with disease risk.

This is the first time that a cohort of laboratory-confirmed Lyme disease cases across England and Wales

has been described in terms of the socioeconomic status of their residential postcode area. The results suggest that patients in England diagnosed with Lyme disease are more likely to live in areas which are more affluent, have high levels of employment and education, have a higher quality of life, are less exposed to crime, but have issues with access to housing and local services. This is in contrast to the classic income gradient of health,[19–21] where the lower an individual's socioeconomic position the worse their health, but supports previous socioeconomic analyses of Lyme disease in the USA.[23 24] This study has not investigated why areas with higher socioeconomic status appear to correlate with a higher incidence of Lyme disease cases but it may reflect the type of leisure activities undertaken, available leisure time, access and attitudes to the countryside by this section of society.[45] Further research is needed to better define the population of cases diagnosed with Lyme disease and why there is an association with socioeconomic status.

The only negative association with Lyme disease in England was observed for the barriers to housing and services domain and is likely due to the rural nature of the areas with the highest incidence. Rural areas score poorly as the housing tends to be expensive in relation to income and houses are a greater distance from services such as hospitals, schools and post offices. It could be reflective of this population only accessing healthcare, and so needing serological diagnosis, once symptoms have progressed beyond the early stages of disease. The living environment deprivation domain is a mix of housing quality, air pollution and road traffic accidents, and it is unsurprising that no association with Lyme disease incidence was observed.

In Wales, there was a significant positive correlation between case counts and the WIMD domain scores. There were an increasing number of patients living in more affluent areas. The reasons for these differences are likely to be similar to the English study population.

The main limitation of this study is the use of patient residence postcode area as a proxy both for the place where Lyme disease was acquired and the socioeconomic status of Lyme disease cases. It is unknown how representative the socioeconomic characteristics of a postcode are of individual cases. Clear socioeconomic and demographic trends and associations have been identified; however, these factors cannot be disentangled using the current data sets and so the degree of bias inherent in them is unknown. Future studies should be designed, where a multivariable model can be created to identify any interaction or confounding effects of the variables under examination.

Current guidance for Lyme disease states that an erythaema migrans rash is pathognomonic and further laboratory diagnostics are not required.[4] An unknown proportion of cases will be clinically diagnosed and managed in early illness by primary care clinicians and will not make it in to this data set. Laboratory-confirmed figures will therefore underestimate the true incidence of Lyme disease seen in the general population. It has been suggested that the underestimate could be between two to six-fold.[13] Without surveillance of primary care presentations, and the use of consistent case definitions and coding, it will be hard to establish a more accurate incidence figure.

The majority of geographical data presented is reliant on case postcode data. Due to data attrition only 58.2% of cases in our data set contained this data. Data attrition may have occurred in three ways: poor completion of the laboratory referral forms (something well documented for health professionals[46]), the non-notifiable status of clinical Lyme disease and the lack of statutory obligation to provide information about suspect cases, and the indirect route by which clinical samples are submitted for testing. Lyme disease testing is usually requested in primary care and samples are routed through hospital laboratories before reaching RIPL. There is the potential that some cases are also missed due to some laboratories (both private and public) performing their own diagnostic testing without sending samples to RIPL, as a specialist diagnostic testing laboratory, for confirmation. Testing rates may also vary in different geographies dependent on Lyme disease awareness of healthcare professionals. The results indicated that the degree of missingness was not even across all PHE regions. This level of missingness had not been anticipated, and there is the potential for bias within the results. It would be possible to extract missing geographical data by linking cases to data sets with patient postcode data, via a unique patient identifier (NHS number). However, data linkage for this data set was not possible as part of public health surveillance under The Health Protection Legislation (England) Guidance 2010.[47] These geographical results should be interpreted within the above context and with an appropriate level of prudence.

In this study it has been shown that laboratory-diagnosed Lyme disease cases in England and Wales have a bimodal age distribution and male predisposition. Geographical clustering of cases was seen in southern England and new insights into the socioeconomics of the resident area of laboratory-confirmed Lyme disease patients were described. This study strengthens the knowledge base of Lyme disease by providing incidence maps which highlight areas where Lyme disease may place the highest burden on primary and secondary care and characterising the socio-demographics of Lyme disease cases. These data will facilitate improved public health interventions and messaging, disease surveillance and patient management.

**Author affiliations**
[1]NIHR Health Protection Research Unit in Emerging and Zoonotic Infections, University of Liverpool, Liverpool, UK
[2]Field Epidemiology Service, Public Health England, Liverpool, UK
[3]Rare and Imported Pathogens Laboratory, Public Health England, Porton Down, UK
[4]NIHR Health Protection Research Unit in Emerging and Zoonotic Infections, Public Health England, Porton Down, UK
[5]National Infection Service, Public Health England, London, UK

[6]Institute of Infection and Global Health, University of Liverpool, Neston, UK
[7]NIHR Health Protection Research Unit in Emerging and Zoonotic Infections, Public Health England, Liverpool, UK

**Acknowledgements** This work uses data provided by patients and collected by the National Health Service as part of their care and support and would not have been possible without access to this data. The National Institute for Health Research recognises and values the role of patient data, securely accessed and stored, both in underpinning and leading to improvements in research and care.

**Contributors** JSPT, RMC, RV, ADR and JCW contributed to the design and implementation of the research. AES, TJGB, KR, KDH and JCW provided the Rare and Imported Pathogens Laboratory data set and assisted in its cleaning. JSPT performed data analysis. JSPT and JCW wrote the manuscript in consultation with KR, KDH, RMC, RV, ADR, TJGB and AES.

**Funding** The research was funded by the National Institute for Health Research (NIHR) Health Protection Research Unit in Emerging and Zoonotic Infections at University of Liverpool in partnership with Public Health England (PHE) and in collaboration with Liverpool School of Tropical Medicine. JSPT, RMC and ADR are based at the University of Liverpool. TJGB, AES and JCW are based at the PHE Rare and Imported Pathogens Laboratory, Porton Down. RV is based at PHE, Liverpool. KR and KDH are based in the Emerging Infection and Zoonoses section at PHE. The views expressed are those of the authors and not necessarily those of the National Health Service, the NIHR and the Department of Health or Public Health England.

**Map disclaimer** The depiction of boundaries on the map(s) in this article do not imply the expression of any opinion whatsoever on the part of BMJ (or any member of its group) concerning the legal status of any country, territory, jurisdiction or area or of its authorities. The map(s) are provided without any warranty of any kind, either express or implied.

**Competing interests** None declared.

**Patient consent for publication** Not required.

**Ethics approval** No ethical approval was required as these anonymised patient data were collected for public health surveillance under The Health Protection Legislation (England) Guidance 2010.

**Provenance and peer review** Not commissioned; externally peer reviewed.

**Data sharing statement** The data used in this research are not made publicly available.

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
