## [Reviewer comments · BMJ Open]

ARTICLE DETAILS

TITLE (PROVISIONAL)	The demographics and geographic distribution of laboratory-confirmed Lyme disease cases in England and Wales (2013-2016): an ecological study
AUTHORS	Tulloch, John; Semper, Amanda; Brooks, Tim; Russell, Katherine; Halsby, Kate; Christley, Robert; Radford, Alan; Vivancos, Roberto; Warner, Jenny

VERSION 1 - REVIEW

REVIEWER	Roger Evans Scottish Lyme Disease and Tick-borne Reference Laboratory, Scotland
REVIEW RETURNED	14-Dec-2018

GENERAL COMMENTS	This is a timely and excellent paper. It does have limitations but these are clearly stated. What is most important is to allow this information to be seen by both healthcare workers and the general public as it addresses a major gap in the current literature of epidemiology of Lyme disease in England and Wales. . I have attached some comments below. The page numbers are as stated on the manuscript itself 'x of 35'. 1. Page 4, line 11. Change 'Ricinus' to 'ricinus'2. Page 4, line 39. Change 'complies' to 'compiles'3. Page 6, line 54. 'duplicates were removed where necessary'. Does this take into account the possibility of re-infection?4. Page 7, line 28/29. Clarify sentence. Either remove 'this was' and begin second half of sentence with 'in this approach, k is defined as the..'5. Page 7, line 54. How many no post codes were excluded?6. Page 8, line 58. Incorrect figure 1. Figure 2 would appear to be figure 1 and vice versa.7. Page 9, line 6. See comment (6) above.8. Page 9, line 17. Only 58.2% of the total study population had an available post code. Were other ways of identifying post codes sought? Was there bias in which post codes were available? For example, were more post codes available from the south of England?
--

	9. Page 15, line 27. Authors state only 56.6% of data contained post code area here but early in manuscript state 58.2%. Why is there a difference in %?
--	--

REVIEWER	Jules Koffi Public Health Agency of Canada
REVIEW RETURNED	16-Dec-2018

GENERAL COMMENTS	Overall the paper is well written. The manuscript presents some interesting and spatial analysis that allow to develop map. The use of the smoothing method to develop the maps seems very helpful. However, the manuscript should be re-reviewed by the authors to improve readability and clarity. Page 6, Line 52-54, among the variables capture, there is "Travel history". Throughout the study, there is no mention about the travel-related cases. It will be worthwhile to present. Page 7 Line 21-36. The sentence "In an attempt to account for the unknown distance..." is too long and doesn't help the comprehension of this part of the method. I suggest to the authors to rewrite this part. The authors could provide the formula used in the k-nearest neighbours (k-NN) approach, even though the reference is provided in the manuscript.t the statistics related to those cases. Page 9, line 31-35: Only four postcode areas had no laboratory-confirmed cases in the four year surveillance period (Fig 3b), could the authors provided some explanations why these 4 post code areas had no laboratory-confirmed cases. Page 11 line 56-59: The authors do not explain why the postal code is available for only 58.2% of the patients and how this affects the spatial analysis. Page 12: line 40-45: the authors made a very speculative affirmation: "These data suggest that although I. ricinus ticks are widespread across England and Wales the proportion that carry B. burgdorferi s.l. is relatively low, and a higher prevalence may only exist in the tick populations in the localities highlighted". This study is not about the prevalence of tick infection by B. burgdorferi, therefore it is not possible to draw a such conclusion. This conclusion can be made only if the study investigated the relationship between the prevalence of infection in tick and the incidence of Lyme disease. I suggest to the authors to delete this sentence or rewrite this sentence and make it clear that can be an explanation to their finding but it has not been tested in this study.
--

VERSION 1 – AUTHOR RESPONSE

Reviewer(s) Reports:

Reviewer: 1

1. Page 4, line 11. Change 'Ricinus' to 'ricinus'

Changed

2. Page 4, line 39. Change 'complies' to 'compiles'

Changed

3. Page 6, line 54. 'duplicates were removed where necessary'. Does this take into account the possibility of re-infection?

This referred to the accidental duplication of a record (so all variables were the same), rather than an individual being re-infected.

Amended to 'duplicate (across all variables) records were removed'

4. Page 7, line 28/29. Clarify sentence. Either remove 'this was' and begin second half of sentence with 'in this approach, k is defined as the..'

'This was' deleted and next sentence begins, 'In this approach.'

5. Page 7, line 54. How many no post codes were excluded?

The following has been added to the results:

'The patient residence postcode was not provided on 1,665 of the referral forms, and therefore only 58.2% (n=2,321) of cases could be described at postcode area resolution.'

6. Page 8, line 58. Incorrect figure 1. Figure 2 would appear to be figure 1 and vice versa.

The figures have been relabelled correctly

7. Page 9, line 6. See comment (6) above.

The figures have been relabelled correctly

8. Page 9, line 17. Only 58.2% of the total study population had an available post code. Were other ways of identifying post codes sought? Was there bias in which post codes were available? For example, were more post codes available from the south of England?

The following has been added to the results:

'The average percentage of missing postcode data by PHE region was 31.9% (range: 10.8%-76.1%). The regions with the highest missing postcode data were London (76.1%), South West (49.4%), and North West (44.7%). The regions with the lowest missing postcode data were Wales (10.8%), North East (12.1%), and West Midlands (14.5%).'

And to the discussion:

'The results indicated that the degree of missingness was not even across all PHE regions. This level of missingness had not been anticipated, and there is the potential for bias within the results. It would be possible to extract missing geographical data by linking cases to datasets with patient postcode data, via a unique patient identifier (NHS Number). However, data linkage for this dataset was not possible as part of public health surveillance under The Health Protection Legislation (England) Guidance 2010.[46] These geographical results should be interpreted within the above context and with an appropriate level of prudence.'

9. Page 15, line 27. Authors state only 56.6% of data contained post code area here but early in manuscript state 58.2%. Why is there a difference in %?

Apologies, this was a typo and should read 58.2%. This has been amended.

Reviewer: 2

Page 6, Line 52-54, among the variables capture, there is "Travel history". Throughout the study, there is no mention about the travel-related cases. It will be worthwhile to present.

The following text has been added in the results;

'Only 10.5% (n=417) of cases had details on the submission form confirming or excluding international travel from a case's clinical history. Due to the low completeness of this variable, it was concluded that further analysis of travel history would not be performed.'

Page 7 Line 21-36. The sentence "In an attempt to account for the unknown distance...." is too long and doesn't help the comprehension of this part of the method. I suggest to the authors to rewrite this part. The authors could provide the formula used in the k-nearest neighbours (k-NN) approach, even though the reference is provided in the manuscript.t the statistics related to those cases.

This has been amended to the below. The authors feel that providing all the formulas needed to describe the k-NN approach would overemphasise this part of the methodology, and distract from the overall focus of the papers. The authors feel that the references used describe this standard smoothing method in enough detail for readers of this article to replicate the methodology as required.

'To account for the unknown distance between a patient's home address and where they were bitten and to highlight any disease hotspots, the disease incidence map for postcode area was smoothed. A k-nearest neighbours (k-NN) approach was used.[28–30] In this approach, a Queen contiguity was used to define geographical neighbours, this defines a neighbour as being an area that shares a common edge or vertex. k is defined as the number of neighbours used for smoothing. k is equal to the square root of the total number of discrete geographical areas rounded to the nearest whole odd number (i.e. 105 postcode areas, its square root being 10.2, therefore k=11). '

Page 9, line 31-35: Only four postcode areas had no laboratory-confirmed cases in the four year surveillance period (Fig 3b), could the authors provided some explanations why these 4 post code areas had no laboratory-confirmed cases.

The following has been added:

'The four postcode areas with no laboratory-confirmed cases were all surrounded by areas with very low incidence and is likely to be reflective of the overall low incidence of Lyme disease in England and Wales.'

Page 11 line 56-59: The authors do not explain why the postal code is available for only 58.2% of the patients and how this affects the spatial analysis.

The reasoning why they were unavailable have been described on page 15, lines 24-51. And the following was added to the discussion:

The results indicated that the degree of missingness was not even across all PHE regions. This level of missingness had not been anticipated, and there is the potential for bias within the results. It would be possible to extract missing geographical data by linking cases to datasets with patient postcode data, via a unique patient identifier (NHS Number). However, data linkage for this dataset was not possible as part of public health surveillance under The Health Protection Legislation (England) Guidance 2010.[46] These geographical results should be interpreted within the above context and with an appropriate level of prudence.

Page 12: line 40-45: the authors made a very speculative affirmation: "These data suggest that although I. ricinus ticks are widespread across England and Wales the proportion that carry B. burgdorferi s.l. is relatively low, and a higher prevalence may only exist in the tick populations in the

localities highlighted”. This study is not about the prevalence of tick infection by *B. burgdorferi*, therefore it is not possible to draw a such conclusion. This conclusion can be made only if the study investigated the relationship between the prevalence of infection in tick and the incidence of Lyme disease. I suggest to the authors to delete this sentence or rewrite this sentence and make it clear that can be an explanation to their finding but it has not been tested in this study.

Rewritten to;

‘Although *I. ricinus* ticks are widespread across England and Wales,[1] the risk of contracting Lyme disease appears to be relatively low. It is possible that the tick populations found within high Lyme disease incidence areas may also have the highest *B.burgdorferi* s.l. prevalence. Several studies would appear to support this hypothesis,[41–43]’

VERSION 2 – REVIEW

REVIEWER	Roger Evans Director, Scottish Lyme Disease and Tick-Borne Reference Laboratory, Raigmore Hospital, Inverness IV2 7DX UK
REVIEW RETURNED	06-Feb-2019

GENERAL COMMENTS	Previous comments have been addressed and the paper reads very well. This is an important piece of work for Lyme disease in England and Wales.
--